# The most exposed regions of SARS-CoV-2 structural proteins are subject to strong positive selection and gene overlap may locally modify this behavior

Alejandro Rubio,[1] Maria de Toro,[2] Antonio J. Pérez-Pulido[1]

**ABSTRACT** The SARS-CoV-2 (severe acute respiratory syndrome coronavirus 2) pandemic that emerged in 2019 has been an unprecedented event in international science, as it has been possible to sequence millions of genomes, tracking their evolution very closely. This has enabled various types of secondary analyses of these genomes, including the measurement of their sequence selection pressure. In this work, we have been able to measure the selective pressure of all the described SARS-CoV-2 genes, even analyzed by sequence regions, and we show how this type of analysis allows us to separate the genes between those subject to positive selection (usually those that code for surface proteins or those exposed to the host immune system) and those subject to negative selection because they require greater conservation of their structure and function. We have also seen that when another gene with an overlapping reading frame appears within a gene sequence, the overlapping sequence between the two genes evolves under a stronger purifying selection than the average of the non-overlapping regions of the main gene. We propose this type of analysis as a useful tool for locating and analyzing all the genes of a viral genome when an adequate number of sequences are available.

**IMPORTANCE** We have analyzed the selection pressure of all severe acute respiratory syndrome coronavirus 2 genes by means of the nonsynonymous (Ka) to synonymous (Ks) substitution rate. We found that protein-coding genes are exposed to strong positive selection, especially in the regions of interaction with other molecules (host receptor and genome of the virus itself). However, overlapping coding regions are more protected and show negative selection. This suggests that this measure could be used to study viral gene function as well as overlapping genes.

**KEYWORDS** SARS-CoV-2, selection pressure, Ka/Ks ratio, overlapping genes

Coronaviruses are positive strand, membrane coated, RNA viruses displaying a complex life cycle that is only partially understood. They belong to order *Nidovirales*, known for presenting the largest found RNA genomes of around 30 kb (1). Coronaviruses enter the host cell by an endocytic mechanism involving the fusion between its coat membrane and the cell plasma membrane. This fusion is mediated by membrane receptor recognition through its spike (S) protein (2–4).

The virus surface also includes M and E proteins that are necessary for efficient assembly, trafficking, and release of virions (5). Also contributing to this is the N protein, which forms the nucleocapsid that binds and packages the virus genome within the virion. Once the nucleocapsid is released into the cytosol, the viral RNA-dependent RNA polymerase is translated using the genomic +RNA molecule as a template and takes over the processes of genome transcription and replication. Apart from these four

Address correspondence to Alejandro Rubio, arubval@upo.es, or Antonio J. Pérez-Pulido, ajperez@upo.es.

The authors declare no conflict of interest.

See the funding table on p. 13.

*[This article was published on 14 December 2023 with an incorrect reference. The References were corrected in the current version, posted on 21 December 2023.]*

structural proteins, two large proteins are translated from the genomic RNA (ORF1a and ORF1ab), which are further proteolytically processed to generate 16 different peptides named Nsp with non-structural functions. Among other functions, they mediate cell membrane reorganization to form double-membrane vesicles where the viral replication and transcription complexes are anchored (6, 7). A complex transcription mechanism synthesizes negative-strand RNA molecules as templates for positive overlapping subgenomic mRNA molecules (8) driving the synthesis of structural and other accessory viral proteins to complete the viral nucleocapsid (6, 9).

Coronaviruses have been known for decades as agents infecting different animal species and causing mild respiratory and gastrointestinal diseases in humans. Recent episodes of epidemic outbreaks causing a large number of fatalities have pointed to the necessity of developing strong therapies against this kind of virus. This has been the case for this latter severe acute respiratory syndrome pandemic, the so-called 2019 novel coronavirus disease (COVID-19), caused by the severe acute respiratory syndrome coronavirus 2 (SARS-CoV-2). Starting at the end of 2019 in the Chinese province of Hubei, it has rapidly expanded throughout the world due to its particularly high infectivity (10).

In less than a year, it rapidly spread globally, having adapted to human hosts, allowing the emergence of thousands of variants subjected to positive selection (11), and some of them have allowed it better fitness. This has led to efforts for logical classification of sequenced genomes. Pango (12), GISAID (13), and Nextstrain (14) are the main developed classification schemes, each of them focusing on differential characteristics of the virus and/or the state of the pandemic. In order to simplify public communication, the World Health Organization Virus Evolution Working Group developed a "Greek letter" system for labeling only a small number of variants of concern (VOC) and variants of interest (15). These variants have an altered phenotype that makes the virus more infectious, pathogenic, or escape natural or acquired immunity. At this time, in October 2022, the pandemic is driven by the omicron lineage, subdivided into multiple lineages that show differential characteristics. Some authors already consider this lineage as a new serogroup of the SARS-CoV-2 (16).

The global SARS-CoV-2 pandemic has occurred in the current genomics era, where whole genome sequencing is a routine activity (17). This has resulted in the availability of an enormous number of genomes from different strains, the comparison of which allows studies of all kinds, more limited with other virus species.

Comparative genomics of this large number of viral genomes now makes it possible to perform studies that could not be considered a few years ago. This can provide information on the genes of the virus, making it possible to study which of them evolve more rapidly, or which of them are fixed in the strains of the virus. For example, like other RNA viruses, SARS-CoV-2 can express small peptides encoded by alternative reading frames of other longer genes, thus overlapping their sequence (18). This means that sequence variants may be affecting two genes at the same time, and therefore it is difficult for the amino acid sequence of both to be maintained. This can be measured by means of the nonsynonymous (Ka) to synonymous (Ks) substitution rate (19). Thus, a Ka/Ks ratio <1 (Ks > Ka) suggests that purifying or negative selection maintains protein structure and function in the corresponding gene, and a Ka/Ks ratio >1 suggests non-purifying selection pressure typically associated with non-coding sequences or positive selection of changes leading to different structures and/or functions in the evolving encoded protein.

SARS-CoV-2 is known to have high positive selection with the emergence of intra-host nonsynonymous mutations, which are subsequently filtered to fix the most relevant for virus fitness (20, 21), and this is something that varies when studying isolated genomes from different geographic regions (22). However, when analyzed as a whole, the entire virus genome shows negative selection, as reflected by its Ka/Ks ratio (23), and most of the observed global positive selection appeared to be concentrated at the beginning of the pandemic (24). In contrast, calculating this ratio with specific genes and specific regions of these genes provides more information on protein domains subject to positive

selection (25), thus highlighting regions of the virus that might interact with the host immune system or those that we need to tailor in vaccination (26). We show the assessment of genome-wide selection pressure, which allows us to study the less-known proteins of the virus, including those that overlap their open reading frames (27). We present, here, the results of an analysis of the selection pressure of the different genes of the virus, including those that have overlaps in their open reading frames. The results show that there is positive selection on the part of the genes encoding the most exposed regions of the corresponding proteins, and that the appearance of overlapping genes interferes with the selection pressure, conserving the overlapping regions to a greater extent.

## RESULTS

### SARS-CoV-2 structural proteins are subject to high positive selection

The early Wuhan strain was used as the reference genome to extract all known SARS-CoV-2 genes, separating them according to whether they were genes encoding structural proteins (S, E, M, and N), non-structural proteins (Nsp1-16), or accessory factors (Fig. 1A). To calculate the selection pressure for each gene, the coding sequence (CDS) of each was used separately (Fig. 1B). Next, the gene was searched in the 1,839 different available viral genomes collected during 8 months in the same region, and the homologous found genes were aligned to the reference CDS, each time using a different reading frame of the six possible ones. This pairwise alignment (the CDS of each genome versus the reference CDS) allowed calculation of the number of synonymous versus nonsynonymous changes, and this allowed calculation of the Ka/Ks ratio, where a value <1 would reflect negative selection and the opposite positive selection. This ratio is more significant (lower *P* value) when the number of orthologs is larger, as well as the number of changes found is higher. Finally, the Ka/Ks ratio values can be compared against the significance of the calculation, with a lower value of the ratio normally expected in frame +1, which is the one encoding the corresponding viral protein. In addition, the

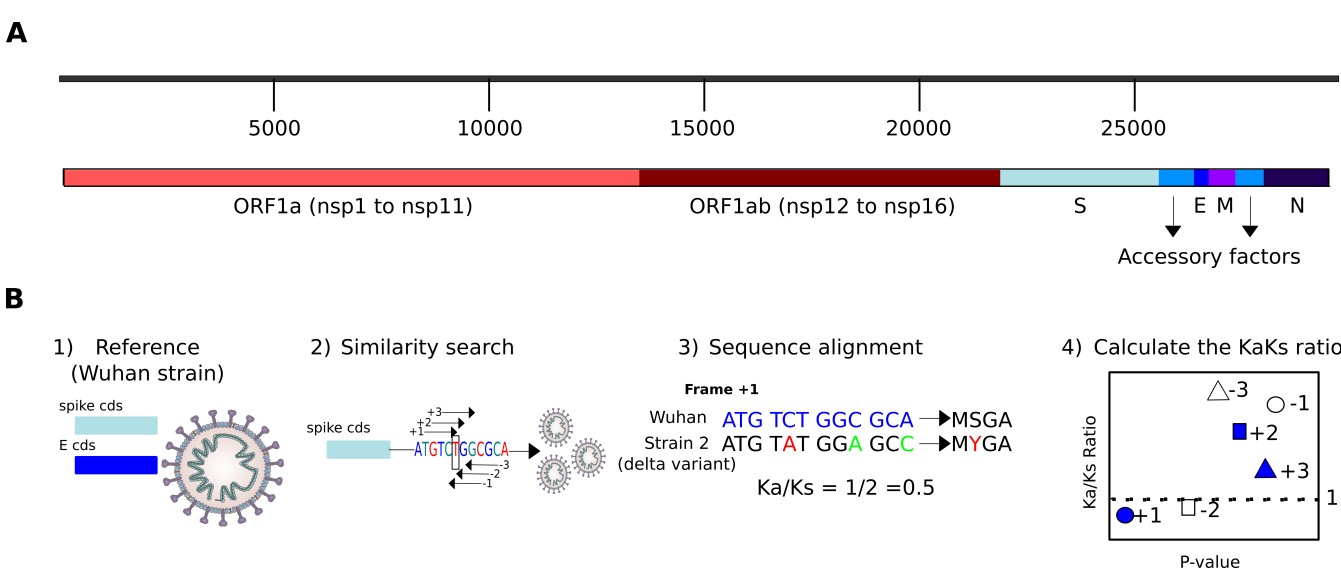

**FIG 1** Viral genes and protocol for calculating the selection pressure. (A) SARS-CoV-2 genome organization. The genome is divided into non-structural genes (*nsp* genes coming from both ORF1a and ORF1ab), structural genes (*S*, *E*, *M*, and *N*), and accessory factors. (B) Procedure for calculating the Ka/Ks ratio: (1) CDSs are obtained from the reference strain (GenBank: MN908947.3). (2) The six putative reading frames for each CDS in the reference strain are extracted, and homologous sequences are searched for in all the strains. (3) The Ka/Ks ratio for each frame can then be calculated using pairwise alignments of the homologs (the same genes from the other viral strains). The analyzed gene is highlighted in blue, and nucleotide changes or in red (when they correspond to nonsynonymous changes), or in green (to synonymous changes). (4) Finally, the distribution of Ka/Ks ratios from all the viral genes can be shown, where we expect a value <1 for most of the genes in the frame +1 (negative selection), and slightly higher values in the frame −2. However, the other four frames should show values >1 (positive selection).

frame −2 shares the third codon position with the frame +1, which means that it is also prone to have a low Ka/Ks ratio (28).

The first thing to note is that the structural genes of the virus show positive selection with a Ka/Ks ratio of around 4.5 (Fig. 2; Table S2 at https://zenodo.org/records/10083993). This result was expected, since they are the proteins most exposed to the host immune system, and they initially interact with proteins of the infected cell (29). However, non-structural proteins, involved in the processes that take place once the virus genome

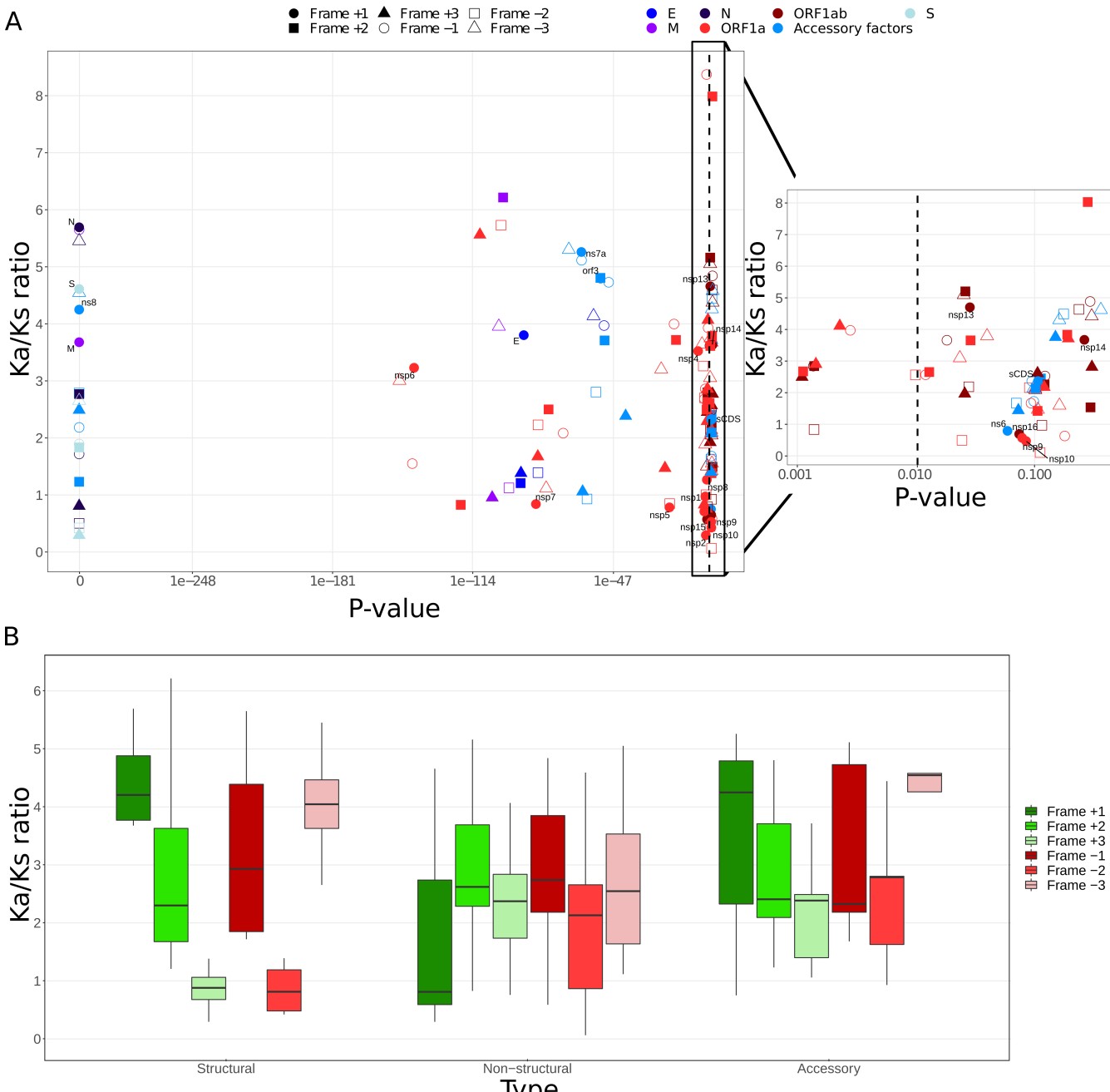

FIG 2   Ka/Ks ratio for SARS-CoV-2 genes. (A) Ka/Ks ratio versus the *P* value obtained (which depends on both number of pairwise alignments used and the length of the sequence) for all SARS-CoV-2 genes. The different shapes of the dots highlight the six possible reading frames, and the different colors highlight the different gene clusters (see legend). The points corresponding to frame +1 are labeled with the gene name. On the right side, a zoom for the range of *P* values between 0.001 and 0.1 has been shown to better see the dispersion. (B) Ka/Ks ratio distribution separated by the six reading frames analyzed and by gene type (structural, non-structural, and accessory).

has entered the cell, show negative selection, which implies that these proteins have essential functions that do not allow major changes in their amino acid sequence. Finally, accessory genes have intermediate values ranging from 0.7 (negative selection) to values >5 (positive selection).

Frame −2 of the non-structural genes also presented low Ka/Ks ratio values (only slightly higher than for frame +1), as expected. However, in the case of structural genes, the ratio of frame −2 was in all cases below 1, suggesting that it would be under high selection pressure, despite not being the frame that encodes the corresponding protein. In fact, the rest of the alternative reading frames mostly have a Ka/Ks ratio less than the coding frame.

When analyzing the proportion of nucleotide substitutions per codon position, in the case of non-structural genes there is a higher rate of changes in the third position, which normally allows the encoded amino acid to be maintained (Fig. 3). However, in the case of structural genes, this rate of changes is more distributed among the three positions.

It should be noted that the Ka/Ks ratio of the structural genes is reliable, since they showed the most significant ratios, with a *P* value of 0. It could be expected that this significance is related to the length of the sequence, since the longer the sequence, the greater the number of changes that could occur. However, when comparing these parameters, there did not seem to be any correlation between them (see Fig. S1 at https://zenodo.org/records/10083993).

## Positive selection is concentrated in exposed regions of structural proteins

Once the selection pressure was calculated at the global level, we wanted to discard that it was randomly distributed along the gene sequence, since it would be expected that there would be regions of the protein that were evolutionarily more protected than others. For this purpose, we calculated the Ka/Ks ratio in sliding windows along the entire gene sequence for the frame +1 (Fig. 4A).

Structural genes of SARS-CoV-2 were used due to their greater Ka/Ks ratio. The *S* gene is the one with a greater divergence in the virus genome, which is something highlighted by its known percentage of mutations per position (Fig. 4B). This gene shows a changing Ka/Ks ratio, which remains practically below 1 in the last 1,000 nucleotides of the gene. Interestingly, the Ka/Ks ratio peaks tended to coincide with the most prevalent mutation sites in the known VOC variants of the virus. However, the region with the highest average Ka/Ks ratio was the receptor binding domain (Pfam:PF09408), which is involved in direct binding to the cellular ACE2 receptor. Furthermore, if we look at the variation

**A**

| | Number of substitutions | | | Percentage | | |
|---|---|---|---|---|---|---|
| | Pos 1 | Pos 2 | Pos 3 | Pos 1 | Pos 2 | Pos 3 |
| **nsp1** | 47 | 54 | 42 | 33% | 38% | 29% |
| **nsp2** | 55 | 47 | 73 | 31% | 27% | 42% |
| **nsp3** | 137 | 138 | 271 | 25% | 25% | 50% |
| **nsp4** | 48 | 33 | 45 | 38% | 26% | 36% |
| **nsp5** | 89 | 74 | 83 | 36% | 30% | 34% |
| **nsp6** | 75 | 92 | 82 | 30% | 37% | 33% |
| **nsp7** | 16 | 28 | 32 | 21% | 37% | 42% |
| **nsp8** | 44 | 47 | 76 | 26% | 28% | 46% |
| **nsp9** | 39 | 40 | 63 | 27% | 28% | 44% |
| **nsp10** | 12 | 20 | 29 | 20% | 33% | 48% |
| **nsp12** | 38 | 92 | 45 | 22% | 53% | 26% |
| **nsp13** | 64 | 60 | 99 | 29% | 27% | 44% |
| **nsp14** | 78 | 56 | 109 | 32% | 23% | 45% |
| **nps15** | 54 | 67 | 51 | 31% | 39% | 30% |
| **nps16** | 60 | 71 | 68 | 30% | 36% | 34% |
| **Total** | 856 | 919 | 1168 | 29% | 31% | 40% |

**B**

| | Number of substitutions | | | Percentage | | |
|---|---|---|---|---|---|---|
| | Pos 1 | Pos 2 | Pos 3 | Pos 1 | Pos 2 | Pos 3 |
| **S** | 53 | 77 | 73 | 26% | 38% | 36% |
| **E** | 11 | 17 | 18 | 24% | 37% | 39% |
| **M** | 60 | 28 | 38 | 48% | 22% | 30% |
| **N** | 148 | 134 | 119 | 37% | 33% | 30% |
| **orf3a** | 92 | 91 | 98 | 33% | 32% | 35% |
| **orf6** | 20 | 20 | 23 | 32% | 32% | 37% |
| **orf7/orf7a** | 47 | 44 | 27 | 40% | 37% | 23% |
| **orf8** | 16 | 34 | 26 | 21% | 45% | 34% |
| **orf10** | 13 | 19 | 17 | 27% | 39% | 35% |
| **Total** | 460 | 464 | 439 | 34% | 34% | 32% |

**FIG 3** Number and percentage of substitutions per codon position of the reading frame +1. Non-structural (A) and structural (B) genes have shown separately. Note that codon position 1 (Pos 1) is the position 1 for frame +1 and position 2 for frame −2, position 2 (Pos 2) is the position 2 for frame +1 and position 1 for frame −2, and position 3 (Pos 3) is common to both frame +1 and −2 [see Fig. 1B(2) for more detail].

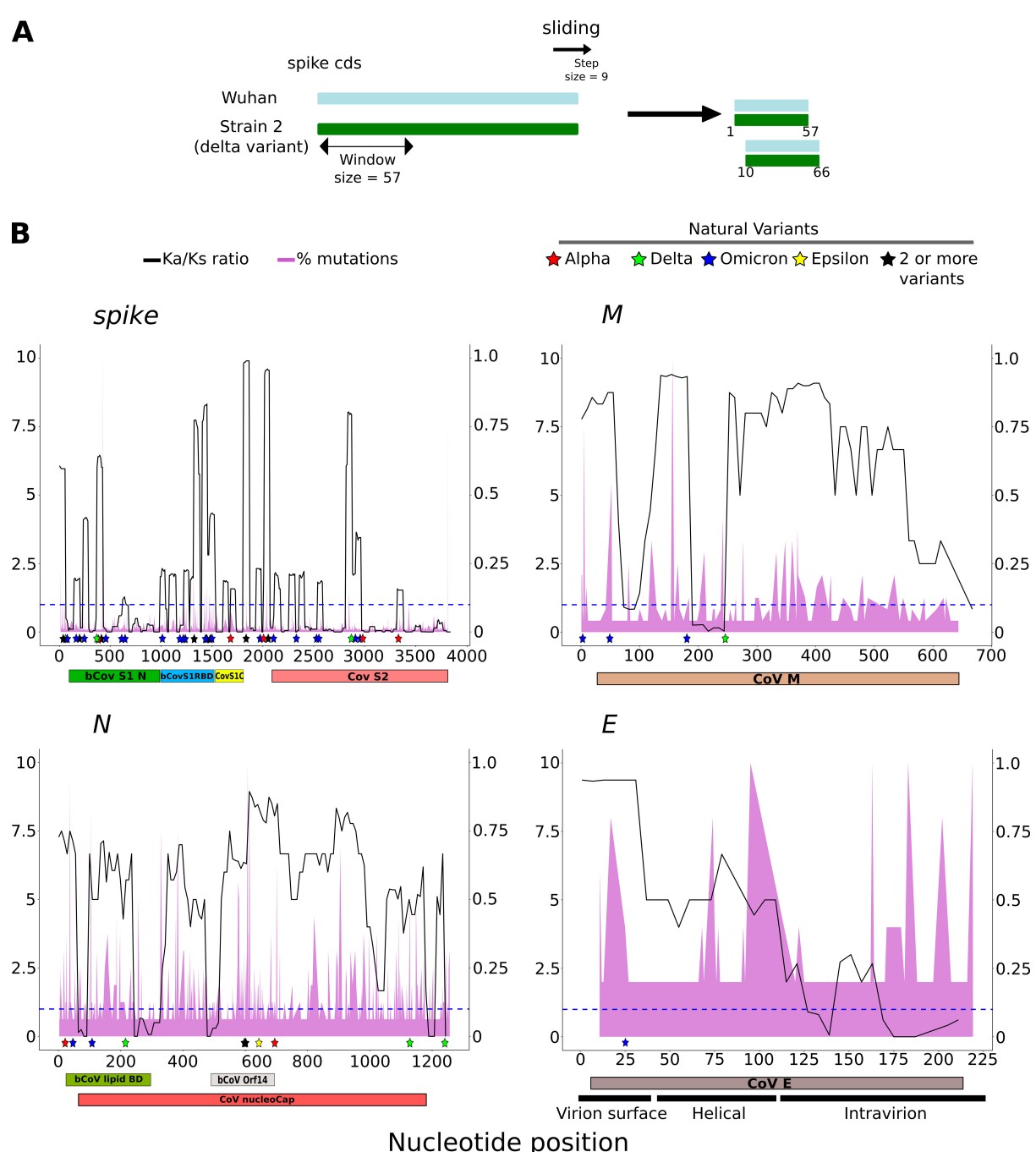

**FIG 4** Ka/Ks ratio distribution along the sequence of structural genes. (A) Ka/Ks ratio is calculated from each pairwise alignment of each gene in a window of 57 nucleotides. The window slides in nine nucleotide steps, and the complete profile is finally plotted along the entire length of the gene. For the E gene, a window of 30 with a slide of 6 was used, due to its short length. (B) Distribution of Ka/Ks along the length of genes *S*, *M*, *N*, and *E* (black line). The percentage of mutations per position obtained from Nextstrain database is also shown for comparison (https://nextstrain.org/ncov/gisaid/global/6m). The primary *Y*-axis represents the Ka/Ks ratio, and the secondary *Y*-axis represents the percentage of mutations in relative value. In addition, VOC from the Outbreak.info database (colored stars), and the Pfam domains have been included (below): S → bCovS1N (PF16451, betacoronavirus-like spike glycoprotein S1, N-terminal), bCoV_S1_RBD (PF09408, betacoronavirus spike glycoprotein S1, receptor binding), CoV_S1_C (PF19209, coronavirus spike glycoprotein S1, C-terminal), CoV_S2 (PF01601, coronavirus spike glycoprotein S2); M → CoVM (PF01635, coronavirus M matrix/glycoprotein); N → bCoV_lipid_BD (PF09399, betacoronavirus lipid binding protein), bCoV_Orf14 (PF17635, betacoronavirus uncharacterized protein 14), CoV_nucleocap (PF00937, coronavirus nucleocapsid); E → CoVE (PF02723, coronavirus small envelope protein E). The blue line marks the Ka/Ks value of 1.

of the Ka/Ks ratio in the structure of protein S, we can see that this domain stands out especially, supporting that it is a very mutable region of the virus (Fig. 5A and B). Finally, it should be noted that the Ka/Ks ratio partly coincides with the Shannon entropy along the gene sequence extracted from the Nextstrain database, called diversity in this database, although the former is more informative (see Fig. S2 at https://zenodo.org/ records/10083993). However, the proportion of changes, marked by the percentage of mutations per position, barely matches the Ka/Ks ratio, which makes the information provided by the latter stand out more. A notable difference appears near nucleotide

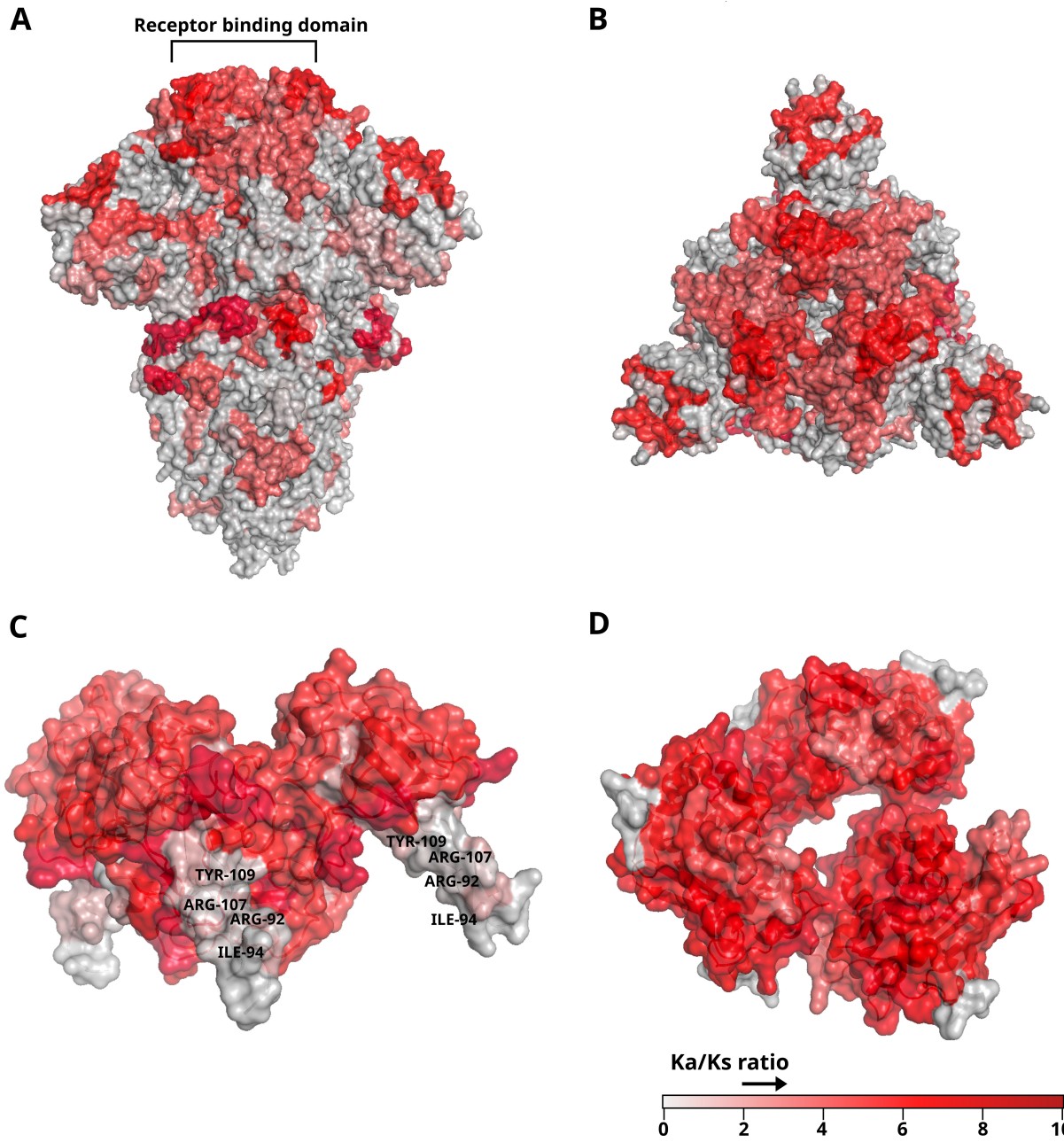

**FIG 5** **S**ARS-CoV-2 spike protein structure highlighting regions with a higher Ka/Ks ratio. (A) Surface and cartoon representation of spike protein (PDB:6VXX). The receptor binding domain has been marked at the top. (B) Receptor binding domain viewed from above. (C) Surface and cartoon representation of the N-terminal region of the N protein (PDB:6M3M, positions 41–174). Amino acids involved in binding to the virus genome, whose mutations are known to affect this binding, have been labeled along with their position in the protein sequence. (D) Surface and cartoon representation of the C-terminal region of the N protein (PDB:6WJI, positions 257–364). All the structures have been colored with different intensities of red depending on the value of the Ka/Ks ratio.

position 500 of the S gene. While Shannon entropy marks it as a highly variable position, the Ka/Ks ratio indicates that there are hardly any amino acid changes.

The other structural genes also showed variations in their Ka/Ks ratio profile but with large regions undergoing positive selection. The *M* gene showed a high level of Ka/Ks in the final two-thirds of its C-terminal end, although no concern variants have been described in this region. The N protein is important for forming the nucleocapsid, and it binds and helps package the virus RNA genome. Thus, its Ka/Ks ratio is high throughout the gene. However, around amino acid position 109 (327 at nucleotide position), the ratio drops to <1 and coincides with a region that decreases RNA binding capacity when changed by mutagenesis (30). This position has a high proportion of changes; however, the low Ka/Ks ratio value reflects that these changes are mainly synonymous. In the 3D structure of this protein, the region in which this position is located is a kind of finger containing other amino acids of special importance for binding to the virus RNA genome (31), which is supported by the low Ka/Ks ratio it presents (Fig. 5C).

On the other hand, the gene *E* encodes a small membrane protein located in the virus envelope (32). It has an N-terminal region located on the surface of the virion, a central transmembrane region, and a C-terminal region that is located inside the virion. When the distribution of the Ka/Ks ratio along the length of this protein was analyzed, it was found that each of the three regions described had a characteristic value (Fig. 4B): the extravirion region seems to be subject to positive selection (the ratio remains constant at a value of 10), the transmembrane region also has values compatible with positive selection, but with lower values (ratio of about 7), and the intravirion region seems to be subject to negative selection, maintaining ratio values <1. All these suggest that the region of the virion exposed to the environment allows for the greatest divergence in this protein.

## Regions of the genome where two genes overlap have higher selection pressure on the main gene

Several genes in the SARS-CoV-2 genome have overlapping genes, whose open reading frame is completely included in the other of the genes already described. Specifically, the structural genes *S* and *N* contain three overlapping genes in frame +2, and *ORF3* contains four overlapping genes, two in frame +2 and two in frame +3 (Fig. 6A). The overlap of these genes means that selective pressure now should affect two genes in the same region of the genome. Therefore, we wanted to compare the Ka/Ks ratio in these overlapping regions, obtaining the value for the overlapping gene and the corresponding region of the gene containing it. Thus, the overlapping gene of the *S* gene (*ORF2b*) showed negative selection despite the positive selection found in the same region of the container gene (Fig. 6B and C). This could partly explain why the Ka/Ks ratio of the −2 reading frame of the *S* gene has a value of 0.4, as opposed to the +1 reading frame which has 4.6 (Fig. 2). On the other hand, the overlapping genes of the *N* gene (*ORF9b* and *ORF9c*) showed high positive selection, with very similar values at the reading frame of the container gene itself, though slightly lower. Finally, two overlapping genes of the gene *ORF3* (*ORF3b* and *ORF3c*) showed negative selection in both the overlapping and the container gene, suggesting that both frames are evolutionarily protected (Fig. 6D).

## DISCUSSION

Measuring selection pressure, by calculating the Ka/Ks ratio, is now possible for complete genomes and is proposed as a very useful system for studying those highly subject to divergence, such as those of viruses and bacteria (28, 33). This makes it possible to classify protein-coding genes into two groups: those subject to negative selection, and therefore more conserved within the species, and those subject to positive selection, usually because they need to constantly modify their sequence.

Here, 1,839 SARS-CoV-2 virus genomes have been analyzed, which were collected by the same hospital over a period of 8 months, allowing the analysis of the selection pressure of their genes in a limited period of time and region. In these genomes,

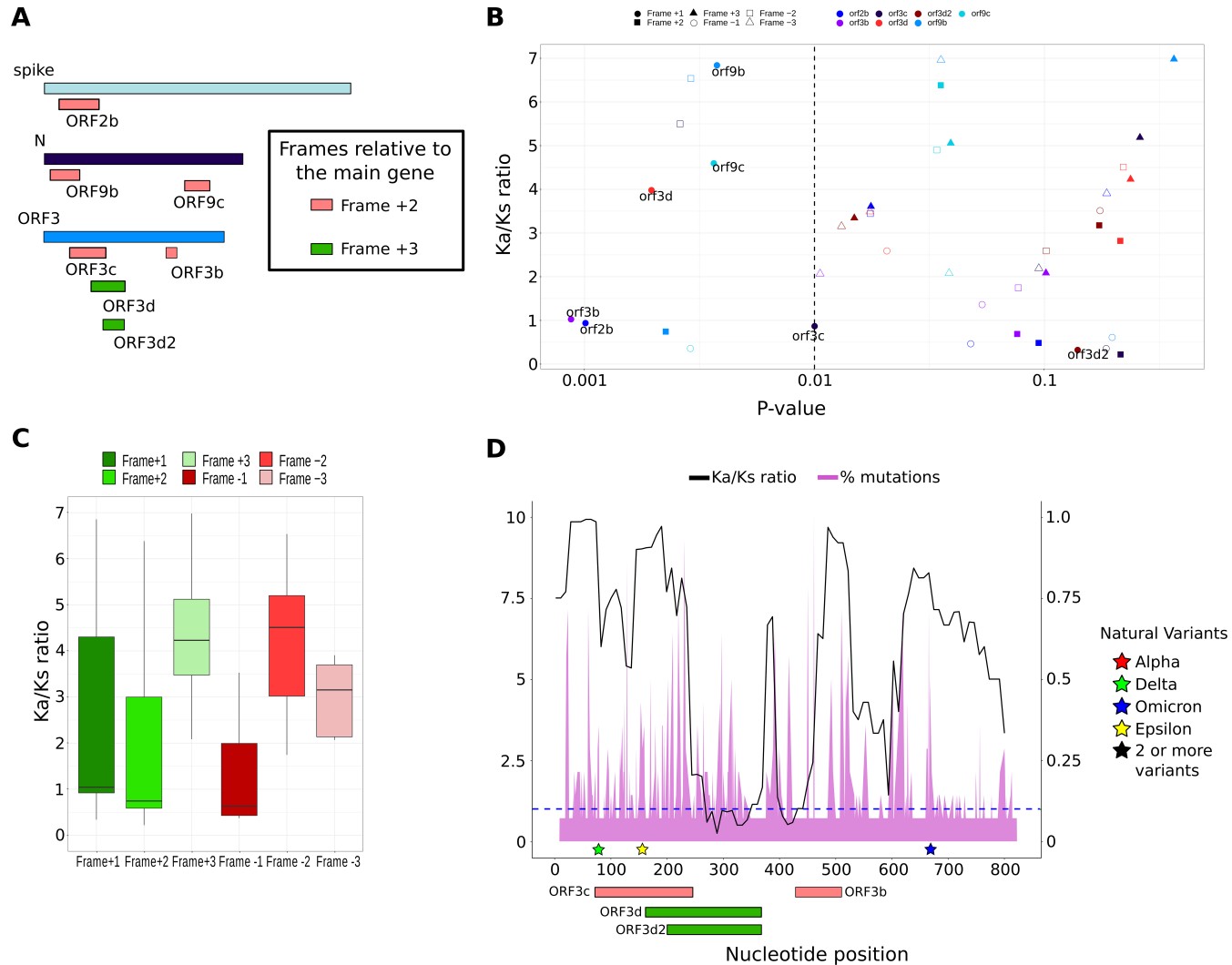

**FIG 6** Ka/Ks ratio in overlapping genes. (A) Overlapping regions of *S*, *N*, and *ORF3* genes. The frame relative to the main gene has been differently colored: +2 (red), +3 (green). (B) Ka/Ks ratio versus *P* value for overlapping ORFs. Genes were distinguished by different colors and frames by different shapes. (C) Ka/Ks ratio distribution separated by the six reading frames analyzed. (D) Distribution of Ka/Ks ratio along the length of gene ORF3. The percentage of mutations per position obtained from Nextstrain database is also shown for comparison (https://nextstrain.org/ncov/gisaid/global/6m). The primary *Y*-axis represents the Ka/Ks ratio, and the secondary *Y*-axis represents the percentage of mutations in relative value. In addition, VOC from the Outbreak.info database were added (colored stars). The blue line marks the Ka/Ks value of 1.

redundant genes were not considered for the calculation of selection pressure, although we can find mutations with a high frequency, but surrounded by other different mutations. This makes these mutations have a higher weight in the analysis, in support of their representativeness in the initial genome data set.

The work presented here is an example of a use that could be useful for selection pressure analysis of other viruses when a large number of genomes are available. In the current genomic era, this number is growing rapidly, allowing for massive evolutionary analyses of viral species (34). However, the rate of change throughout the entire SARS-CoV-2 pandemic has not been assessed here, as has been evaluated in other works (35). The fact that all genomes used here are genomes processed in the same way guarantees an unbiased homogenization of the data set used. This virus has a total of 31 genes, some of which constitute a set that is under selection pressure to retain its core functions, and other genes that must continually adapt to the environment to overcome the host immune system and cellular defense systems (36). These genes subjected to

positive selection, as occurs in bacteria, coincide with genes that encode for surface proteins, since these are the ones that are in contact with the environment. In the case of SARS-CoV-2, we found positive selection exclusively in structural proteins *S*, *E*, *M*, and *N*, with non-structural proteins showing negative selection, because they are normally subject to strong selection pressure. This high rate of positive selection has not been found in the related SARS-CoV virus, which 16 years earlier caused an epidemic that led to hundreds of deaths and hospitalizations (37). However, the small size of that epidemic, together with the low number of genomes that could be sequenced, could explain this difference.

The value of 1 in the Ka/Ks ratio, which represents an equal number of synonymous and nonsynonymous changes, can mark a threshold between negative and positive selection. When we measure the selection pressure of a gene, it is done on the open reading frame encoding its corresponding protein, the +1 reading frame (20, 23, 25). Since this is the reading frame to be conserved, the other five alternative reading frames usually have a higher Ka/Ks ratio than the main one. However, mainly in structural genes, alternative reading frames have shown a lower Ka/Ks ratio than the main one in our study, suggesting that evolution is conserving them. This fact has been unexpected and it would be necessary to study other viral genomes, to know if it is something preserved, since this is not the case in bacteria (28). Genes with a high Ka/Ks ratio may represent either spurious genes or real genes that are under positive selection. So, perhaps, the low value appearing in the alternative reading frames could be pointing to genes from the second group mentioned, as suggested by the results presented here. Thus, the low Ka/Ks ratio of the alternative reading frames could maintain the positive selection of the main frame.

The measurement of the percentage of mutations along the genome sequence, and their relationship to the reading frame of genes, is currently helpful in assessing the selection pressure of these genes (21, 38). Here, we show that the calculation of the Ka/Ks ratio in the six possible ORFs of the genes can help to classify them. Thus, in our study, we found that the non-structural genes present selection pressure in the +1 and −2 reading frames, which have in common the last codon position. However, structural genes show positive selection in the +1 reading frame, while in the −2 reading frame selection pressure is maintained, something that seems to define this group of genes. Finally, the accessory genes seem to present an intermediate behavior.

Selection pressure may vary along the sequence of a given gene. For example, in the case of genes encoding membrane proteins, the extracellular region may be subject to positive selection, but the unexposed region may not. This has been verified here in the case of the structural *S* and *E* genes. In the case of the gene coding for the nucleocapsid protein, which is bound to the virus genome and is exposed to the cell's defense systems once it has entered the cell, it shows positive selection, with Ka/Ks ratio values near 10 along its entire length. However, around nucleotide position 327 (109 in amino acids), the ratio drops to values close to 0. Site-directed mutagenesis experiments have been carried out in this region, and the change of the amino acid at positions ARG92, ILE94, ARG107, and TYR109 produces a significant loss of binding capacity to the RNA genome of the virus (30, 31). This suggests that knowledge of these regions on essential pathogenic viruses and bacteria could be of great interest as drug and vaccine targets, since they represent constant regions of the gene or regions that are not subject to divergence, such as the variants of concerns in SARS-CoV-2 (11).

Other indicators used to study the variability of the virus genome are the proportion of changes and the Shannon entropy, which is an index of diversity by position, given by the Nextstrain database (14). However, these measures do not help to know whether the changes are evolutionarily allowed and affect the coding sequence. The latter is something that Ka/Ks ratio analysis does allow, helping to propose sites for targeted mutagenesis experiments or epitopes for vaccines.

Since protein-coding genes are usually subject to negative selection, the main reading frame is the one that normally has a lower Ka/Ks ratio. This makes regions of

a gene that have a lower Ka/Ks ratio in alternative reading frames suggestive of hosting new overlapping genes. It has been tested here with the overlapping genes known in SARS-CoV-2. Furthermore, these overlapping reading frames could partly explain the low Ka/Ks ratio values that structural genes exhibit in alternative reading frames with respect to the value given by their reading frame +1. The analysis of the Ka/Ks ratio in single-stranded RNA viruses may even be useful to find or validate genes encoded in the complementary strand of the virus genome, something that in SARS-CoV-2 has already been predicted by means of a computational procedure, in which the 3D structure and possible function of these non-experimentally validated genes have been studied (39). Ka/Ks ratio analysis could be useful to highlight these regions with possible alternative coding, as we previously demonstrated in bacteria (28). Thus, new unknown genes in viral genomes could be proposed for further laboratory validation.

The results presented here show how to analyze the selection pressure to which the genes of a viral genome are subjected, which is not only useful for locating highly conserved regions and drug targets, but also allows the analysis of overlapping genes or genes with low support. This methodology and results represent a proof of concept, which will allow performing the same type of analysis with other viruses in the future, allowing to analyze groups of specific strains, genes of special interest to locate the most evolutionarily protected regions, and to evaluate the prediction of new genes, as well as to classify them.

## MATERIALS AND METHODS

### Samples

All samples ($N = 1,839$) originate from patients positively diagnosed with COVID-19 at Hospital Universitario San Pedro (Logroño, Spain), who provided the total RNA used in the diagnosis for the whole genome sequencing and research purposes. Samples were randomly recruited between July 2021 and March 2022 and sequenced in the Center for Biomedical Research of La Rioja (CIBIR), and they range from variant of concern Alpha to variant Omicron (see Table S1 at https://zenodo.org/records/10083993). Both the laboratory and bioinformatics protocols were agreed upon and standardized in the national genomic epidemiology network established to track SARS-CoV-2 variants, as described previously (40).

### cDNA synthesis, library preparation, and sequencing

All Next-Generation Sequencing (NGS) libraries were prepared in sets of 96 samples. The first step comprised the cDNA synthesis from the initial total RNA. In this step, the viral RNA is denatured and reverse transcribed to cDNA to serve as a template in subsequent PCR amplification steps. At the second step, the complete viral genome is amplified from the cDNA obtained in the previous step by means of two multiplex PCRs, by using the ARTIC v4 primers sets described at https://github.com/artic-network/artic-ncov2019/tree/master/primer_schemes/nCoV-2019/V4.

After mixing the two amplicon pools, this DNA is cleaned by using a bead-based system and quantified using a fluorometric procedure (Qubit 3.0, Life Technologies). All these steps are extracted from the original ARTIC protocol (Quick 2020).

Latest steps comprise the library preparation by following the Illumina DNA Prep Tagmentation protocol (Illumina, San Diego, CA, USA), consisting of fragmenting the amplified viral cDNA, followed by a cleaning step and the amplification of the fragmented DNA and indexing. Finally, each library is quantified by using a Qubit 3.0 fluorometer (Life Technologies) and qualified by capillary electrophoresis in Fragment Analyzer (Agilent Technologies). Libraries from each preparation are pooled based on the quality control evaluation at equimolar amounts of 4 nM and sequenced at a final concentration of 7.5 pM (1% PhiX at 7.5 pM) on the MiSeq platform using the MiSeq Reagent Kit v2 (300 cycle) (Illumina, San Diego, CA, USA).

## Reference mapping, variant discovery, and lineage designation

The sequences obtained were analyzed through a bioinformatic pipeline designed by the SeqCOVID Consortium (https://seqcovid.csic.es/), available at https://gitlab.com/fisabio-ngs/sars-cov2-mapping and modified to adapt to our server. Briefly, the pipeline goes through the following steps: (i) quality control prior to trimming with FastQC v0.11.9 (41); (ii) removal of the human reads with Kraken v2.0.8 (42); (iii) low quality, short reads, and adapters trimmed by fastp 0.20.0 (43) (arguments: -cut_tail, -cut-win-dow_size 10, -cut_mean_quality 30); and (iv) mapping and variant calling using bwa 0.7.17-r1188 (44) and iVar v1.3.1 (45) against the MN908947.3 reference genome (variant calling cut-offs: minimum quality for Single Nucleotide Polymorphism (SNP) calling = 20, minimum frequency to call an SNP = 0.05, minimum depth for calling an SNP = 30 and consensus construction cut-offs: minimum quality for consensus calling = 20, minimum frequency to consider fixed an SNP = 0.8, minimum position depth = 30, and ambiguous base otherwise).

Finally, consensus sequences were classified into SARS-CoV-2 lineages (12) by using the Pangolin software (46) at the following versions: pangolin v3.1.20; pangoLEARN v2022-02-28; constellations v0.1.3; scorpio v0.3.16, and pango-designation v1.2.132. All determined sequences were submitted to GISAID database and are included under the EPI_SET_221102kp accession number with https://doi.org/10.55876/gis8.221102kp (13).

## Ka/Ks ratio calculation

We used the protocol proposed in our previous work to calculate the Ka/Ks ratio using the software KaKs_Calculator 2.0 with some modifications (28, 47). The CDS of the reference strain (Wuhan-Hu-1, GenBank identifier: MN908947.3) (48) was used as the starting input, and BLASTN 2.9.0+ (49) was used to search for homologous sequences in all available different strains sequenced. Homologous sequences were aligned in pairs with the reference CDS when they were found, using MAFFT version 7.305 with the starting sequences (50). When the homologous gene is not found in a given genome, with 90% identity and 95% coverage, it was not considered for the analysis, which helps to eliminate poor-quality sequences. In addition, genes with exactly the same nucleotide sequence were not included in the alignment to avoid redundancy. When gaps are not multiples of three in one of the sequences, then the missing gaps are added to the end of the sequence. The sequences of these alignments are then transformed into the five remaining reading frames by Seqkit software version 0.15. Finally, the Ka/Ks ratio for the six putative reading frames is calculated with the pairwise alignment as input for KaKs_Calculator, and the mean values of Ka/Ks ratio and $P$ value (Fisher's exact test) were collected.

To calculate the Ka/Ks ratio in a site-specific manner, a sliding window analysis along the CDS genes was designed using an in-house PERL script. This splits each global alignment into small alignments of between 30 and 57 nucleotides in length with an offset of between 6 and 9 nucleotides, as shown in reference (47). The Ka/Ks ratio for the coding frame is then calculated with the pairwise alignments as input to KaKs_Calculator. Finally, the distribution of the ratio along each gene is plotted, and functional domains described in InterPro 89.0 (51) are added. In addition, the ratio of mutations per position and diversity (normalized Shannon entropy) extracted from Nextstrain (14) is incorporated.

The number of changes per codon position for reading frame +1 has been calculated by counting the number of unique changes of all available genomes against the reference strain genome.

## 3D structure analysis

To analyze how the Ka/Ks ratio varies along the 3D structure, the crystal structure corresponding to the proteins (S and N) was obtained from the PDB database (PDB identifiers 6VXX, 6M3M, and 6WJI) (52) and visualized using PyMOL. Subsequently, the

Ka/Ks gradient obtained with sliding window analysis was created, marking the positive selection starting from the value of 1.

## ACKNOWLEDGMENTS

We would like to thank C3UPO for the HPC support. We also want to thank to Laboratorio de Microbiología (Hospital Universitario San Pedro, Logroño, Spain), Maria Pilar Bea Escudero (CIBIR, La Rioja, Spain) and to the SeqCOVID consortium for the support on collecting, sequencing, and analyzing the SARS-CoV-2 genomes included in this paper. We would like to thank Alex Bateman for helpful comments on the manuscript.

This methodology developed for this research has been funded in part by PID2020-114861GB-I00/AEI/10.13039/501100011033 (Agencia Estatal de Investigación/Ministry of Science and Innovation of the Spanish Government).

## AUTHOR AFFILIATIONS

[1]Faculty of Experimental Sciences, Genetics Area, Andalusian Centre for Developmental Biology (CABD, UPO-CSIC-JA), University Pablo de Olavide, Sevilla, Spain
[2]Genomics and Bioinformatics Core Facility, Center for Biomedical Research of La Rioja, Logroño, Spain

## AUTHOR ORCIDs

Alejandro Rubio  http://orcid.org/0000-0002-6736-6141
Maria de Toro  http://orcid.org/0000-0003-3329-0203
Antonio J. Pérez-Pulido  http://orcid.org/0000-0003-3343-2822

## FUNDING

| Funder | Grant(s) | Author(s) |
| --- | --- | --- |
| Ministerio de Ciencia e Innovación (MCIN) | PID2020-114861GB-I00 | Antonio J. Pérez-Pulido |

## DATA AVAILABILITY

All data generated or analyzed during this study are included in this published article and its supplementary information files. The code used is available in the following GitHub repository: https://github.com/UPOBioinfo/KaKsSARS-CoV-2. Supplementary files can be found at https://zenodo.org/records/10083993. The genomes used are deposited in GISAID (EPI_SET_221102kp).

## ADDITIONAL FILES

The following material is available online.

### Open Peer Review

**PEER REVIEW HISTORY (review-history.pdf).** An accounting of the reviewer comments and feedback.

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
