## [Reviewer comments · mSystems]

The most exposed regions of SARS-CoV-2 structural proteins are subject to strong positive selection and gene overlap may locally modify this behavior

Alejandro Rubio, Maria de Toro, and Antonio Pérez-Pulido

Corresponding Author(s): Antonio Pérez-Pulido, Universidad Pablo de Olavide

Review Timeline:

Submission Date:	July 10, 2023
Editorial Decision:	November 2, 2023
Revision Received:	November 8, 2023
Accepted:	November 10, 2023

Editor: Irina El Khoury

Reviewer(s): Disclosure of reviewer identity is with reference to reviewer comments included in decision letter(s). The following individuals involved in review of your submission have agreed to reveal their identity: Nash D Rochman (Reviewer #1)

Transaction Report:

DOI: <https://doi.org/10.1128/msystems.00713-23>

Re: mSystems00713-23 (The most exposed regions of SARS-CoV-2 structural proteins are subject to strong positive selection and gene overlap may locally modify this behavior)

Dear Dr. Antonio J Pérez-Pulido:

Please address the comments by the reviewer #1. I would also consider changing the background in Figure 5 from black to white (might need to recolor the protein domains for visibility).

Revision Guidelines

Sincerely,
Irina El Khoury
Editor
mSystems

Reviewer #1 (Comments for the Author):

The authors present an interesting, timely study charting variation in selection pressures across the SARS-CoV-2 genome by computing the K_a/K_s ratio over sliding windows. Positive selection is demonstrated to be strongest in motifs of the genome which code for exposed regions of structural proteins. Nonstructural proteins are demonstrated to primarily evolve under

purifying selection. Perhaps the most interesting result is the demonstration that regions of the genome corresponding to more than one overlapping ORF are subject to relatively increased purifying selection. While these results may not be wholly unexpected, I find the clear demonstration of these trends in the manuscript compelling. I have two comments.

First, the authors compute Ka/Ks over all possible reading frames in their demonstration of the effects of known ORF overlap. The authors also briefly discuss how unexplained variation in Ka/Ks over alternative reading frames may indicate the presence of additional, unidentified ORFs. Along similar lines, the authors have previously applied these methods to novel gene identification (<https://academic.oup.com/bib/article/23/2/bbac010/6519794#338439246>). There has been some discussion of the possibility of negative-sense ORFs in SARS-CoV-2 (<https://academic.oup.com/bib/article/23/3/bbac045/6539840>) and perhaps more broadly among other (+)ssRNA viruses. I'm curious to know if the authors have considered how these methods could be used to identify such cases.

Secondly, the authors focus on only 1839 SARS-CoV-2 genomes collected over an 8 month period from a single clinical center. I hope the authors could motivate why these genomes were studied specifically instead of working with a much larger sample of available sequences. I do not believe the results will substantially change with the inclusion of a greater number of sequences but I do feel an explanation of the rationale to include only this relatively small dataset is important. Furthermore, the authors suggest the number of SARS-CoV-2 genomes analyzed enables, "studies of all kinds, which are not possible with other virus species." While there are certainly few viruses with the same magnitude of data availability as SARS-CoV-2 (all Influenza), there are several more with more than 2k complete, non-redundant genomes available. Specifically, studies detailing variation in selection pressures across the genomes of these viruses, as is the focus of this manuscript, have been completed (I will selfishly direct the authors to consider my own as an example: <https://www.pnas.org/doi/abs/10.1073/pnas.2121335119>).

I restate that I believe this to be an interesting and timely study.

Sincerely,

Nash Rochman (invited to review 07/24/23; review returned to editor 07/25/23)

“The most exposed regions of SARS-CoV-2 structural proteins are subject to strong positive selection and gene overlap may locally modify this behavior”

Response to reviewers

We would like to thank Dr. Nash Rochman for the review of our article and the words about it. His work has undoubtedly made the article better and we have now thought of new ideas about the project that we had not thought of before.

Reviewer #1 (Comments for the Author):

The authors present an interesting, timely study charting variation in selection pressures across the SARS-CoV-2 genome by computing the Ka/Ks ratio over sliding windows. Positive selection is demonstrated to be strongest in motifs of the genome which code for exposed regions of structural proteins. Nonstructural proteins are demonstrated to primarily evolve under purifying selection. Perhaps the most interesting result is the demonstration that regions of the genome corresponding to more than one overlapping ORF are subject to relatively increased purifying selection. While these results may not be wholly unexpected, I find the clear demonstration of these trends in the manuscript compelling. I have two comments.

First, the authors compute Ka/Ks over all possible reading frames in their demonstration of the effects of known ORF overlap. The authors also briefly discuss how unexplained variation in Ka/Ks over alternative reading frames may indicate the presence of additional, unidentified ORFs. Along similar lines, the authors have previously applied these methods to novel gene identification (<https://academic.oup.com/bib/article/23/2/bbac010/6519794#338439246>). There has been some discussion of the possibility of negative-sense ORFs in SARS-CoV-2 (<https://academic.oup.com/bib/article/23/3/bbac045/6539840>) and perhaps more broadly among other (+)ssRNA viruses. I'm curious to know if the authors have considered how these methods could be used to identify such cases.

This seems to us to be a very interesting possibility for this type of analysis. In fact, it was as a result of our article applied to the search and validation of "missing genes" that Dr. Alex Bateman contacted us. Upon reading our work, it seemed to him that it might be something of interest for searching for spurious proteins in databases, and validating computational predictions. And so, we have explored some ways to do this in collaboration with him.

In the current manuscript we mention the work of Bartas et al., but certainly we do not discuss much the potential of the Ka/Ks calculation in gene prediction. Therefore, we have now added the following paragraph in the discussion:

“The analysis of the Ka/Ks ratio in single-stranded RNA viruses may even be useful to find or validate genes encoded in the complementary strand of the virus genome, something that in SARS-CoV-2 has already been predicted by means of a computational procedure, in which the 3D structure and possible function of these non-experimentally validated genes have been studied (38). **Ka/Ks ratio analysis could be useful to highlight these regions with possible**

alternative coding, as we previously demonstrated in bacteria (28). Thus, new unknown genes in viral genomes could be proposed for further laboratory validation.”

Secondly, the authors focus on only 1839 SARS-CoV-2 genomes collected over an 8 month period from a single clinical center. I hope the authors could motivate why these genomes were studied specifically instead of working with a much larger sample of available sequences. I do not believe the results will substantially change with the inclusion of a greater number of sequences but I do feel an explanation of the rationale to include only this relatively small dataset is important. Furthermore, the authors suggest the number of SARS-CoV-2 genomes analyzed enables, "studies of all kinds, which are not possible with other virus species." While there are certainly few viruses with the same magnitude of data availability as SARS-CoV-2 (all Influenza), there are several more with more than 2k complete, non-redundant genomes available. Specifically, studies detailing variation in selection pressures across the genomes of these viruses, as is the focus of this manuscript, have been completed (I will selfishly direct the authors to consider my own as an example):

<https://www.pnas.org/doi/abs/10.1073/pnas.2121335119>.

It is true that this fact could appear to be a handicap of our study. However, our initial idea was always to use homogeneous and comparable data, as we emphasize in several parts of the article:

- “Here, 1839 SARS-CoV-2 virus genomes have been analyzed, which were collected by the same hospital during a period of 8 months, allowing the analysis of the selection pressure of their genes in a limited period of time and region.”
- “The fact that all genomes used here are genomes processed in the same way guarantees an unbiased homogenization of the data set used.”

These are genomes that have been processed by the co-author of the paper, Dr. Maria de Toro, and we wanted to establish a collaboration with her, to publish and analyze her own data, which were uniform.

In fact, we have already raised the possibility of analyzing other RNA virus species, more with the idea of analyzing the unexplained result that occurs on the complementary strand (the selection pressure in the -2 reading frame). But we would need time and means to carry it out. In fact, I make here the proposal for a possible collaboration.

To be concrete, we have modified the sentence in the introduction in order to be more precise:

“This has resulted in the availability of an enormous number of genomes from different strains, the comparison of which allows studies of all kinds, **more limited** with other virus species.”

In the discussion, we have included a new sentence, making it clear that indeed genomes of other viruses are already available in current databases and evolutionary studies can be carried out:

“The work presented here is an example of a use that could be useful for selection pressure analysis of other viruses, when a large number of genomes are available. **In the current genomic era, this number is growing rapidly, allowing for massive evolutionary analyses of viral species (34).** However, the rate of change throughout the entire **SARS-CoV-2** pandemic has not been assessed here, as has been evaluated in other works (35).”

I restate that I believe this to be an interesting and timely study.

Sincerely,

Nash Rochman (invited to review 07/24/23; review returned to editor 07/25/23)

Thank you again for all the time spent reviewing our work and congratulations on the work you do.

Re: mSystems00713-23R1 (The most exposed regions of SARS-CoV-2 structural proteins are subject to strong positive selection and gene overlap may locally modify this behavior)

Dear Dr. Antonio J Pérez-Pulido:

Your manuscript has been accepted, and I am forwarding it to the ASM production staff for publication. Your paper will first be checked to make sure all elements meet the technical requirements. ASM staff will contact you if anything needs to be revised before copyediting and production can begin. Otherwise, you will be notified when your proofs are ready to be viewed.

Featured Image Submissions: If you would like to submit a potential Featured Image, please email a file and a short legend to mSystems@asmusa.org. Please note that we can only consider images that (i) the authors created or own and (ii) have not been previously published. By submitting, you agree that the image can be used under the same terms as the published article. File requirements: square dimensions (4" x 4"), 300 dpi resolution, RGB colorspace, TIF file format.

Sincerely,
Irina El Khoury
Editor
mSystems